# Auto Learning Attention

**Benteng Ma**[1,2]*     **Jing Zhang**[2]*     **Yong Xia**[1,3]     **Dacheng Tao**[2]

[1]Northwestern Polytechnical University, China
[2]The University of Sydney, Australia
[3]Research & Development Institute of Northwestern Polytechnical University, Shenzhen
{mabenteng@mail,yxia@}.nwpu.edu.cn
{jing.zhang1, dacheng.tao}@sydney.edu.au

## Abstract

Attention modules have been demonstrated effective in strengthening the representation ability of a neural network via reweighting spatial or channel features or stacking both operations sequentially. However, designing the structures of different attention operations requires a bulk of computation and extensive expertise. In this paper, we devise an Auto Learning Attention (AutoLA) method, which is the first attempt on automatic attention design. Specifically, we define a novel attention module named high order group attention (HOGA) as a directed acyclic graph (DAG) where each group represents a node, and each edge represents an operation of heterogeneous attentions. A typical HOGA architecture can be searched automatically via the differential AutoLA method within 1 GPU day using the ResNet-20 backbone on CIFAR10. Further, the searched attention module can generalize to various backbones as a plug-and-play component and outperforms popular manually designed channel and spatial attentions for many vision tasks, including image classification on CIFAR100 and ImageNet, object detection and human keypoint detection on COCO dataset. Code is available at https://github.com/btma48/AutoLA.

## 1   Introduction

Attention learning has been increasingly incorporated into convolutional neural networks (CNNs) [1], aiming to compact the image representation and strengthen its discriminatory power [2, 3, 4, 5]. It has been widely recognized that attention learning is beneficial for many computer vision tasks, such as image classification, segmentation, and object detection.

There are two types of typical attention mechanisms. The channel attention is able to reinforce the informative channels and to suppress irrelevant channels of feature maps [2], while the spatial attention enables CNNs to dynamically concentrate processing resources at the location of interest, resulting in better and more effective processing of information [4]. Let either the channel attention or spatial attention be treated as the first-order attention. The combination of both channel attention and spatial attention constitutes the second-order attention, which has been proven in benchmarks to produce better performance than either first-order attention by modulating the feature maps in both channel-wise and spatial-wise [4]. Accordingly, we propose to extend attention modules from the first- or second-order to a higher order, i.e., arranging more basic attention units structurally. However, considering the highly variable structures and hyperparameters of basic attention units, exhaustively searching the architecture of high order attention module is an exponential explosion in complexity.

Recent years have witnessed the unprecedented success of neural architecture search (NAS) in the automated design of neural network architectures, surpassing human designs on various tasks [6, 7, 8, 9, 10, 11, 12]. We advocate the use of NAS to search the optimal architecture of high order attention module, which however is challenging for several reasons. First, there is no explicit off-the-shelf definition of the search space for attention modules, where various attention operations may be included [13]. Second, the sequential structure for arranging different attention operations should be computationally efficient so that it can be searched within affordable computational budgets. Third, how to search the attention module, e.g., given the backbone or together with the backbone cells, remains unclear. Fourth, the searched attention module is expected to generalize well to various backbones and tasks.

In this paper, we propose an Auto Learning Attention (AutoLA) method for automatically searching efficient and effective plug-and-play attention modules for various well-established backbone networks. Specifically, we first define a novel concept of attention module, i.e., high order group attention (HOGA), by exploring a 'split-reciprocate-aggregate' strategy. Technically, each HOGA block receives feature tensor from each block in the backbone as input, which is divided into $K$ groups along the channel dimension to reduce the computational complexity. Then, a directed acyclic graph (DAG) [14] is constructed, where each node is associated with a split group, and each edge represents a specific attention operation. The sequential connections between different nodes can represent different combinations of basic attention operations, resulting in various first-order to $K$th order attention modules, which indeed constitute a search space of HOGA. By customizing DARTS [8] for our problem, the explicit HOGA structure can be searched efficiently within 1 GPU day on a modern GPU given a fixed backbone network (e.g., ResNet-20) on CIFAR10. Extensive experiments demonstrate the obtained HOGA generalizes well on various backbones and outperforms previous hand-crafted attentions for many vision tasks, including image classification on the CIFAR100 and ImageNet datasets, object detection, and human keypoint detection on the COCO dataset.

To summarize, the contribution of our paper is three-fold. First, to the best of our knowledge, AutoLA is the first attempt to extend NAS to search plug-and-play attention modules beyond the backbone architecture. Second, we define a novel concept of attention module named HOGA that can represent high order attentions and the previous channel attention and spatial attention can be treated as its special cases. Third, we utilize a differentiable search method to search the optimal HOGA module efficiently, which can generalize well on various backbones and outperform previous attention modules for many vision tasks.

## 2 Related work

**Attention mechanism.** The attention mechanism was originally introduced in neural machine translation to handle long-range dependencies [15], which enables the model to attend to important regions within a context adaptively. Self-attention was added to CNNs by either using channel attention or non-local relationships across the image [2, 3, 16, 17, 18]. As different feature channels encode different semantic concepts, the squeeze-and-excitation (SE) attention captures channel correlations by selectively modulating the scale of channels [2, 19]. Spatial attention is also explored together with the channel attention in [4], resulting in a second-order attention module called CBAM and achieving superior performance. In [19, 20], the attention is extended to multiple independent branches which achieves improved performance than the original one. In contrast to these hand-crafted attention modules, we define the high order group attention and construct the search space accordingly where SE [2] and CBAM [4] are special instances in it. Consequently, a more effective attention module can be searched automatically, outperforming both SE [2] and CBAM [4] on various vision tasks.

**Neural Architecture Search.** In terms of the NAS methods, reinforcement learning [9, 21, 22], sequential optimization [10, 23], evolutionary algorithms [24, 25, 26], random search [27, 28], and performance predictors [29, 30] tend to demand immense computational resources which probably not suitable for efficient search. Recent NAS methods reduce the search time significantly by weight-sharing [14, 31, 32] and continuous relaxation of the space [8, 33]. DARTS [8] and its variants [34, 35, 36] only need to train a single neural network with repeated cells during the searching process [37], providing elegant differentiable solutions to optimizing network weights and architecture simultaneously. Besides, DARTS is computationally efficient which is only slightly slower than training one architecture in the search space. Instead of searching basic cells and stacking

them sequentially to form the backbone network, we propose to search an efficient and plug-and-play attention module given a fixed backbone, aiming to enhance the representation capacity of the backbone. The searched attention module shows good generalization ability for various backbones and downstream tasks, implying that the proposed method could be complementary to existing NAS-based search of the backbone architectures. Note that the architectures of backbone and attention module can be searched alternatively or synchronously in a framework that we leave as future work.

# 3 Auto Learning Attention

## 3.1 High Order Group Attention

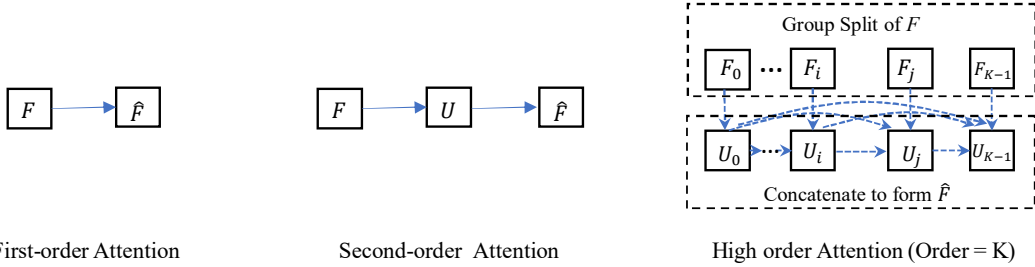

First-order Attention &emsp;&emsp;&emsp; Second-order Attention &emsp;&emsp;&emsp; High order Attention (Order = K)

Figure 1: Attention Order. The typical channel attention, spatial attention, and normalization attention are all first-order attentions. CBAM is a second-order attention. The solid line represents the specific attention operation and the dotted line in High order Attention represents the candidate attention operation which will be searched automatically.

From the view of computational flow, attention operation represents a function that transforms the input tensor $F$ to an enhanced representation $\hat{F}$ through a series of attention operations. This can be formalized in the computational graph, where the operations are represented as a directed acyclic graph with a set of nodes $U$. Each node $U_k$ represents a tensor (we use the same symbol to represent the node and the tensor at the node without causing ambiguity). An attention operation $o \in \mathcal{O}$ is defined on the edge between $U_k$ and its parent nodes $P_k$. In the typical first-order attention module, each node has a single parent and $|P_k| = 1$. Denoting the parent feature tensor $P_k \in R^{C \times H \times W}$ as input, the above attention operation can be defined as:

$$U_k = o\left(P_k\right). \tag{1}$$

Obviously, the increased order of attention may increase the computational complexity. To generate an efficient high order attention module, we divide the input tensor $F$ into $K$ groups along the channel dimension, where $K$ is a cardinally hyper-parameter. In this case, we get $F = \{F_0, F_1, ..., F_{K-1}\}$ which is illustrated in Figure 1. Then, a series of operations $o \in \mathcal{O}$ (where $\mathcal{O}$ is the search space which will be further explained in section 3.2) are applied on the split group features $\{F_i\}_{i=0}^{K-1}$. This process generates $K$ intermediate features as shown in Figure 1, where the $k$th intermediate tensor is calculated as:

$$U_k = o_{k,k}F_k + \sum_{j<k} o_{j,k}U_j, \tag{2}$$

where $U_k \in R^{H \times W \times C/K}$ for $k \in 0, 1, ..., K-1$, $H$, $W$ and $C$ are the sizes of the block output feature map, $o_{i,j} \in \mathcal{O}$ denotes the attention operation between node $U_i$ and $U_j$. Specifically, $o_{k,k}$ represents the attention operation applied from $F_k$ to $U_k$. These intermediate tensors are concatenated along the channel axis to generate the final attentive output:

$$\hat{U} = \varphi[U_0; U_1; ...; U_{K-1}, \overline{U}], \tag{3}$$

where $\overline{U} = \sum_{i=0}^{K-1} U_i$, $[;]$ denotes the concatenation along the channel axis, and $\varphi$ denotes the mapping function learned by $1 \times 1$ convolutional layers and channel attention.

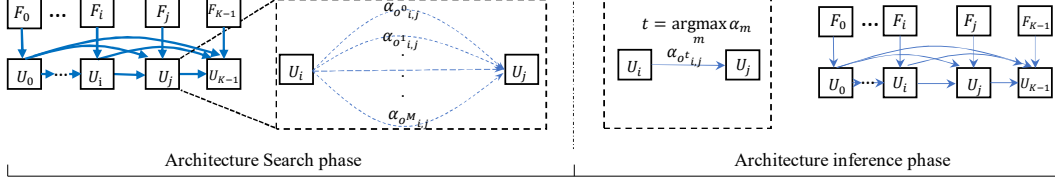

Architecture Search phase            Architecture inference phase

Figure 2: HOGA architecture search. The left represents the searching stage of HOGA architecture and the line between each two nodes represents the mixed attention operations with architecture weight $\alpha_{o_{i,j}}$ and the right represents the searched HOGA architecture which is a sampled sub-graph of the full graph used in the searching stage.

## 3.2 Attention Searching Space

Given the structure of HOGA defined above, searching an HOGA module means sampling an attention operation from a search space for each $o_{j,k}$ in Eq. (2). Here we propose an attention search space which includes typical attention operations used in the literature. We also include some operations such as identity mapping as attention operation in our search space for a unified terminology. Details are presented as follows.

**(a) Channel Attention:** Following [2], we define the channel attention as:

$$A_c\left(F\right) = \mathcal{S}\left(f_c\left(F_{avg}^c\right)\right) \otimes F, \tag{4}$$

where $F_{avg}^c \in R^{C \times 1 \times 1}$ represents the global spatial average pooled feature from the input $F$, $f_c$ is a Multilayer perceptron (MLP), and $\mathcal{S}$ denotes the Sigmoid activation function.

**(b) Channel Attention v2:** In contrast to the channel attention defined in Eq. (4), a variant of it is to utilize both average pooling and max pooling when calculating the pooled feature, i.e.,:

$$A_{c2}\left(F\right) = \mathcal{S}\left(f_c\left(F_{avg}^c\right) + f_c\left(F_{max}^c\right)\right) \otimes F, \tag{5}$$

where $F_{max}^c$ is the max pooled feature. Note that $F_{ave}^c$ and $F_{max}^c$ are fed into the same MLP $f_c$.

**(c) Spatial Attention:** Similar to the channel attention v2, we define the spatial attention as:

$$A_s\left(F\right) = \mathcal{S}\left(f_s\left(\left[F_{avg}^s; F_{max}^s\right]\right)\right) \otimes F, \tag{6}$$

where $F_{avg}^s$ and $F_{max}^s$ represent the global spatial average and max pooled feature, respectively. $[;]$ denotes the concatenation operator. $f_s$ represents a convolutional layer. In this paper, we use a depth-wise convolutional layer with a $3 \times 3$ kernel and a dilation rate chosen from $\{1, 2\}$ to improve the computational efficiency of the attention module.

**(d) Normalization Attention:** Normalization attention refers to normalizing a tensor to the scale within $[0, 1]$ and using it at the attention map, i.e.,

$$A_n\left(F\right) = \mathcal{S}\left(f_n\left(F\right)\right) \otimes F, \tag{7}$$

where $f_n$ is a depth-wise convolutional layer like $f_s$ in Eq. (6).

**(e) Convolutional Block Attention Module (CBAM):** We also include the CBAM in our search space, which can be treated as a combination of the channel attention v2 and spatial attention, i.e.,:

$$A_b(F) = A_s\left(A_{c2}\left(F\right)\right). \tag{8}$$

**(f) Identity and Zero Attention:** Specifically, we also name two special attention operations to represent the identity mapping and zero mapping, respectively, i.e.,

$$A_k(F) = \mathbf{k} \otimes F, \mathbf{k} \in \{\mathbf{1}, \mathbf{0}\}. \tag{9}$$

## 3.3 HOGA Architecture Search

With the comprehensive coverage of the search space and the definition of HOGA, we can formulate the problem of HOGA architecture search. The ultimate objective of the searching is to find an

optimized HOGA architecture that minimizes the expected loss $\mathcal{L}$. We denote the datasets as $D$, the attention search space as $\mathcal{O}$ and the space of all candidate HOGA architectures as $\mathcal{H}$. A general architecture search algorithm is defined as the following mapping:

$$\Gamma : \mathcal{D} \times \mathcal{O} \rightarrow \mathcal{H}. \tag{10}$$

Given a specific dataset $d$, which is split into a training partition $d_{train}$ and a validation partition $d_{val}$, the searching algorithm estimates the model $h_{\alpha,\theta} \in \mathcal{H}_\alpha$, where $\alpha$ is the architecture parameter of the HOGA and $\theta$ is the learnable weight of the model. The best HOGA architecture is searched by minimizing a loss function $\mathcal{L}$ as follows:

$$\alpha^* = \underset{\alpha}{\operatorname{argmin}} \, \mathcal{L}(\Gamma(\alpha, d_{train}), d_{val}). \tag{11}$$

As shown in Figure 2, the HOGA can be represented as a directed acyclic graph (DAG): $G = (V, E)$. Each node $U_i \in V, i = 0, 1, ..K - 1$, represents an intermediate tensor, and the corresponding edge $e_{i,j} \in E$ represents a candidate attention operation which is predefined in the searching space $\mathcal{O}$. Let $o_{i,j} = \{o_{i,j}^0, o_{i,j}^1, ..., o_{i,j}^{M-1}\}$ be the set of candidate attention operations on edge $e_{i,j}$, where $M = |\mathcal{O}|$. In each HOGA module, we assume that each operation outputting $U_k$ only receives a single input from a former node. Accordingly, the intermediate tensor in HOGA is obtained by:

$$U_k = \sum_{m \in |\mathcal{O}|} o_{k,k}^m (F_k | \beta_{k,k,m} = 1) + \sum_{i < k} \sum_{m \in |\mathcal{O}|} o_{i,k}^m (U_i | \beta_{i,k,m} = 1), \tag{12}$$

where $\beta_{i,j,m} \in \{0, 1\}, m \in \{0, 1, ..., M - 1\}$, denoting whether the attention operation $o_{i,j}^m$ is applied. Since $\mathcal{O}$ is a discrete set, similar to [8], we introduce a continuous relaxation to make the search space of HOGA continuous so that the architecture can be optimized through gradient descent. We relaxes the attention categorical choice to a softmax over all operations in $\mathcal{O}$ to form a fusion output:

$$\hat{o}_{i,j}(U_i) = \sum_{o \in \mathcal{O}} \frac{exp(\alpha_{o_{i,j}})}{\sum_{o' \in \mathcal{O}} exp(\alpha_{o'_{i,j}})} o(U_i), \tag{13}$$

where $\alpha_{o_{i,j}}$ denotes the corresponding architecture weight for the attention operation $o_{i,j}$ on $e_{i,j}$. And a discrete architecture can be obtained by replacing each mixed operation $\hat{o}_{i,j}$ with the most likely operation and we adopt the the following approximation to generate the optimal discrete attention architecture:

$$softmax(\alpha_{o_{i,j}}) \rightarrow \beta_{i,j}, \tag{14}$$

where $\beta_{i,j} = (\beta_{i,j,1}, \beta_{i,j,2}, ..., \beta_{i,j,M})$.

The attention architecture search problem is reduced to learn $\alpha^*$ and the network weight $\omega^*$ that minimize the validation loss $\mathcal{L}_{val}(\omega^*, \alpha^*)$, which can be solved by the bi-level optimization:

$$\begin{aligned} \min_{\alpha} \, &\mathcal{L}_{val}(\omega^*(\alpha), \alpha) \\ s.t. \, \omega^*(\alpha) = &\underset{\omega}{\operatorname{argmin}} \, \mathcal{L}_{train}(\omega, \alpha) \end{aligned} \tag{15}$$

To solve Eq. (15), we adopt the first-order approximation in [8] and partition the training data into two disjoint sets $train_A$ and $train_B$. Then we optimize both the weights of the network and architecture parameters of the HOGA by alternating gradient descent, i.e.,

- Train the network weights $\omega$ by $\mathcal{L}_{trainA}(\omega, \alpha)$,
- Update HOGA architecture weights $\alpha$ by $\mathcal{L}_{trainB}(\omega, \alpha)$,

where the loss function $\mathcal{L}$ is the cross entropy calculated on the mini-batch. The partition of training data is to prevent the architecture from overfitting the train data. As shown in Figure 2, the candidate attention operations at each edge are mixed with the normalized continuous coefficient $\alpha_{o_{i,j}}$ during the training phase. When finishing the training, we replace the mixed attention for edge $e_{i,j}$ to a single operation. Thus $e_{i,j}$ can be encoded into a one-hot vector where

$$\beta_{i,j,m} = \begin{cases} 1, & m = \operatorname{argmin}_k \alpha_{o_{i,j}^k} \\ 0, & \text{othervise} \end{cases} \tag{16}$$

Consequently, the specific attention operation can be determined for each edge.

Table 1: Comparison of different attention modules on CIFAR10.

|  | Acc(%) | Param.(M) | FLOPS(G) |
|---|---|---|---|
| ResNet20 [1] | 91.95 | 0.27 | 0.04 |
| ResNet20 + SE [2] | 92.30 | 0.29 | 0.04 |
| ResNet20 + CBAM [4] | 92.81 | 0.30 | 0.04 |
| ResNet20 + AutoLA | **93.38** | 0.34 | 0.05 |
| ResNet32 [1] | 92.55 | 0.46 | 0.07 |
| ResNet32 + SE [2] | 93.16 | 0.49 | 0.07 |
| ResNet32 + CBAM [4] | 93.47 | 0.49 | 0.07 |
| ResNet32 + AutoLA | **94.33** | 0.51 | 0.09 |
| ResNet56 [1] | 93.03 | 0.85 | 0.13 |
| ResNet56 + SE [2] | 94.02 | 0.90 | 0.13 |
| ResNet56 + CBAM [4] | 94.10 | 0.92 | 0.13 |
| ResNet56 + AutoLA | **94.78** | 1.04 | 0.16 |

Table 2: Comparison of different attention modules on CIFAR100

|  | Acc(%). | Param.(M) | FLOPS(G) |
|---|---|---|---|
| ResNet20 [1] | 75.42 | 4.07 | 0.65 |
| ResNet20 + SE [2] | 76.84 | 4.13 | 0.65 |
| ResNet20 + CBAM [4] | 76.93 | 4.14 | 0.67 |
| ResNet20 + AutoLA | **77.85** | 5.23 | 0.71 |
| ResNet32 [1] | 75.72 | 6.85 | 1.10 |
| ResNet32 + SE [2] | 77.81 | 6.97 | 1.10 |
| ResNet32 + CBAM [4] | 78.01 | 7.01 | 1.10 |
| ResNet32 + AutoLA | **78.57** | 8.91 | 1.30 |
| ResNet56 [1] | 77.56 | 12.41 | 2.01 |
| ResNet56 + SE [2] | 79.05 | 13.84 | 2.01 |
| ResNet56 + CBAM [4] | 79.07 | 13.85 | 2.02 |
| ResNet56 + AutoLA | **79.59** | 14.48 | 2.37 |

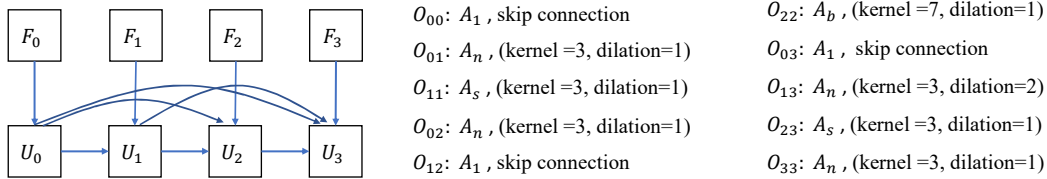

$O_{00}$: $A_1$ , skip connection

$O_{01}$: $A_n$ , (kernel =3, dilation=1)

$O_{11}$: $A_s$ , (kernel =3, dilation=1)

$O_{02}$: $A_n$ , (kernel =3, dilation=1)

$O_{12}$: $A_1$ , skip connection

$O_{22}$: $A_b$ , (kernel =7, dilation=1)

$O_{03}$: $A_1$ , skip connection

$O_{13}$: $A_n$ , (kernel =3, dilation=2)

$O_{23}$: $A_s$ , (kernel =3, dilation=1)

$O_{33}$: $A_n$ , (kernel =3, dilation=1)

Figure 3: The searched architecture of HOGA.

## 4 Experiments

### 4.1 Datasets

Four benchmark datasets, including CIFAR10 [38], CIFAR100 [38], ImageNet ILSVRC2012 [39], and COCO [40], are used for this study.

### 4.2 Experiment Setup

HOGA is a general module which can be integrated into any well-established CNN architectures and is end-to-end trainable along with the backbone. Taking ResNet20 [1] as an exemplar backbone network, where the base number of channel (width) is 16, we search the best architecture of the attention module on it and then transfer the searched attention module to ResNet-32 and ResNet-56 for the evaluation on CIFAR10 and CIFAR100. To evaluate the capabilities of image classification on larger datasets, we transfer the searched attention module to ResNet-18, ResNet-34, ResNet-50, ResNet101 [1], and WiderResNet [41] and train them on ImageNet. When testing on CIFAR100 and ImageNet, the base channel number of the network is set to 64. To further evaluate the generalization capability [42] of the searched HOGA, we incorporate it into ResNet50, pretrain the network on ImageNet, and apply the pretrained network to two heavy downstream tasks, i.e., object detection and human keypoint detection on the COCO dataset.

In the training stage, we set the order of HOGA to $K = 4$ to achieve a trade-off between accuracy and complexity. We randomly split the training set of CIFAR10 into two parts evenly, one for tuning network parameters (denoting $train_A$) and the other one for tuning the attention architecture (denoting $train_B$). The architecture search procedure is conducted for a total of 100 epochs with a batch size of 128. When training network weights $\omega$, we adopt the SGD optimizer with a momentum 0.9 and a weight decay 0.0003, and the cosine learning rate policy that decays from 0.025 to 0.001 [43]. The initial value of $\alpha$ before softmax is sampled from a standard Gaussian and times 0.001. In the evaluation stage, the standard test set is used.

### 4.3 Image Classification Results on CIFAR10 and CIFAR100

In the evaluation stage on CIFAR10, the entire training set is used, and the network is trained from scratch for 500 epochs with a batch size of 256. The results are summarized in Table 1 and Table 2, which demonstrate the searched HOGA attention (denoting "AutoLA") outperforms other attention baselines with slightly more computations. Figure 3 shows the searched architecture of HOGA. As

Table 3: Comparison of different attention modules on ResNeXt and PNAS.

|  | Acc (%) | Param (M) | FLOPS (G) |  | Acc (%) | Param (M) | FLOPS (G) |
|---|---|---|---|---|---|---|---|
| ResNext | 94.76 | 1.71 | 0.28 | PNAS | 93.34 | 0.72 | 0.08 |
| ResNext + SE | 95.22 | 2.23 | 0.30 | PNAS + SE | 93.71 | 0.75 | 0.08 |
| ResNext + CBAM | 95.31 | 2.24 | 0.31 | PNAS + CBAM | 93.79 | 0.76 | 0.08 |
| ResNext + AutoLA | **95.67** | 2.35 | 0.41 | PNAS + AutoLA | **94.10** | 0.91 | 0.11 |

can be seen, compared to the hand-crafted attention modules, the searched HOGA contains more connections between various types of attentions. We also presented the results for other different backbones, including ResNeXt [44] and the one searched by PNAS [23] on CIFAR10 in Table 3. From Table 1, Table 2, and Table 3, we can see that the HOGA searched by AutoLA outperforms other attention modules on CIFAR10 when deployed on highly variable architectures including ResNet, ResNeXt, and PNAS, indicating the consistent superiority of the HOGA searched by AutoLA over previous attention methods.

## 4.4 Image Classification Results on ImageNet

We also perform image classification on the ImageNet dataset to evaluate the searched HOGA module for this more challenging task. We adopt the same data augmentation scheme as [4, 8] for training. We plug the searched HOGA module from the above experiment into various backbone networks including ResNet18, ResNet34, ResNet50, ResNet101 [1], and WideResNet [41]. More training details are in the supplementary material.

Table 4 summarizes the results on the validation set of ImageNet. As can be seen, CBAM [4] marginally outperforms SE [2] on shallow backbones such as ResNet18, and ResNet34. By contrast, it achieves better performance than SE by a larger margin on the WiderResNet, ResNet50, and ResNet101 backbones. We suspect that the second-order attention benefits from more diverse and representative features from the deeper back-

Table 4: Comparison of different attentions on ImageNet

|  | Top-1 Error (%) | Param. (M) | FLOPS. (G) |
|---|---|---|---|
| ResNet18 [1] | 29.60 | 11.69 | 1.81 |
| ResNet18 + SE [2] | 29.41 | 11.78 | 1.82 |
| ResNet18 + CBAM [4] | 29.27 | 12.01 | 1.82 |
| ResNet18 + AutoLA | **27.90** | 13.51 | 2.10 |
| WResNet18 [41] | 26.85 | 26.85 | 3.87 |
| WResNet18 + SE [2] | 26.21 | 26.07 | 3.87 |
| WResNet18 + CBAM [4] | 26.10 | 26.08 | 3.89 |
| WResNet18 + AutoLA | **25.02** | 29.76 | 4.55 |
| ResNet34 [1] | 26.69 | 21.80 | 3.67 |
| ResNet34 + SE [2] | 26.13 | 21.96 | 3.67 |
| ResNet34 + CBAM [4] | 25.99 | 21.96 | 3.67 |
| ResNet34 + AutoLA | **24.65** | 24.63 | 4.29 |
| ResNet50 [1] | 24.56 | 25.56 | 4.11 |
| ResNet50 + SE [2] | 23.14 | 28.09 | 4.12 |
| ResNet50 + CBAM [4] | 22.66 | 28.09 | 4.12 |
| ResNet50 + AutoLA | **21.82** | 29.39 | 4.73 |
| ResNet101 [1] | 23.38 | 44.55 | 7.57 |
| ResNet101 + SE [2] | 22.35 | 49.33 | 7.58 |
| ResNet101 + CBAM [4] | 21.51 | 49.33 | 7.58 |
| ResNet101 + AutoLA | **20.95** | 51.81 | 8.94 |

bones. As for the proposed AutoLA, it outperforms all baselines by large margins on all backbones. We present the explanations as follows. Firstly, AutoLA searches the optimal architecture of HOGA from numerous candidates in the search space via a differentiable algorithm. It has already surpassed many candidates which may include some first- and second-order attentions. Secondly, due to the diverse combinations and transformations in HOGA, it can learn more representative features after a series of feature mapping steps even for shallow backbone networks.

Further, we also compare the performance with other recent well-designed attention modules, including GENet [3], GCNet[45], and AugAtt [17] where GC-Net and AugAtte are non-local attentions. The results are listed in Table 5. All this models are based on ResNet50 backbone with different attention modules. With comparable or even less parameters and

Table 5: Results of other attentions modules on ImageNet.

|  | Top-1 Error (%) | Param (M) | FLOPS (G) |
|---|---|---|---|
| ResNet50 + GENet [3] | 22.00 | 31.20 | 3.87 |
| ResNet50 + AugAtt [17] | 22.30 | 24.30 | 7.90 |
| ResNet50 + GCNet [45] | 22.30 | 28.08 | 3.87 |
| ResNet50 + AutoLA | **21.82** | 29.39 | 4.73 |

FLOPS, the proposed AutoLA outperforms other attention modules by a substantial margin.

## 4.5 FLOPS Fair Comparison and Ablation Study on Image Classification

To further analyse the performance of AutoLA, we increase the width of the backbone networks for SE and CBAM (denoted by "Wide") and further customize SE and CBAM using the group split operation (denoted by "HOG"), resulting in a specific instantiation of HOGA (i.e., k=4) in which all the operations in HOGA are SE/CBAM attentions in these two cases and the FLOPS are fair for the AutoLA. The results on CIFAR10 are listed in Table 6. It reveals that HOGA searched by AutoLA (k=4)) still outperforms SE and CBAM by a large margin. And the expansion backbone with SE and CBAM even contain more parameters than AutoLA (k=4), which confirms the superiority of the proposed AutoLA.

We also present the ablation study on the number of group split (i.e., the hyper-parameter K). From Table 6, less groups mean lower order of attentions in HOGA, leading to inferior performance. If we set $K = 8$ or other larger number, the parameters and FLOSP would increase a lot, thus we take $K = 4$ in our final setting. We also test the generalization ability of HOGA searched on ResNet56 (denoted by "AutoLA_56") on a new backbone, i.e., ResNet20. Although the results are inferior to the ones searched directly on ResNet20, this HOGA still outperforms SE and CBAM. We also compare the attention modules generated by random search and AutoLA in Table 6. The HOGA searched by AutoLA outperforms its randomly searched counterparts (denoted by "Rand"). Note that the attention modules by random search exceed SE and CBAM.

Table 6: Experiments with fair settings of parameters and FLOPGs and ablation study results on CIFAR10.

| | Acc (%) | Param (M) | FLOPS (G) | | Acc (%) | Param (M) | FLOPS (G) |
|---|---|---|---|---|---|---|---|
| ResNet20 + SE | 92.30 | 0.29 | 0.04 | ResNet32 + SE | 93.16 | 0.49 | 0.07 |
| ResNet20 + CBAM | 92.81 | 0.3 | 0.04 | ResNet32 + CBAM | 93.47 | 0.49 | 0.07 |
| ResNet20_Wide + SE | 93.16 | 0.36 | 0.05 | ResNet32_Wide_SE | 94.08 | 0.62 | 0.09 |
| ResNet20_Wide + CBAM | 93.13 | 0.37 | 0.05 | ResNet32_Wide_CBAM | 93.92 | 0.63 | 0.09 |
| ResNet20 + HOG_SE (k=4) | 92.87 | 0.32 | 0.05 | ResNet32 + HOG_SE (k=4) | 93.62 | 0.54 | 0.09 |
| ResNet20 + HOG_CBAM (k=4) | 93.07 | 0.35 | 0.05 | ResNet32 + HOG_CBAM (k=4) | 93.72 | 0.56 | 0.09 |
| ResNet20 + AutoLA (k=2) | 93.18 | 0.33 | 0.05 | ResNet32 + AutoLA (k=2) | 93.81 | 0.49 | 0.09 |
| ResNet20 + AutoLA_56 (k=4) | 93.31 | 0.35 | 0.05 | ResNet32 + AutoLA_56 (k=4) | 94.18 | 0.57 | 0.09 |
| ResNet20 + Rand_HOGA (k=4) | 93.28 | 0.35 | 0.05 | ResNet32 + Rand_HOGA (k=4) | 94.15 | 0.59 | 0.09 |
| ResNet20 + AutoLA (k=4) | 93.38 | 0.34 | 0.05 | ResNet32 + AutoLA (k=4) | 94.33 | 0.52 | 0.09 |

## 4.6 Object Detection Results on COCO

Image classification networks provide generic image features that may be transferred to other computer vision tasks. As an example, we evaluate the usefulness of the searched HOGA module for object detection in this part. Specifically, we choose the popular object detection framework named Single-Shot Detector (SSD) [46] and a popular two-stage framework Faster RCNN [47] + FPN [48] use ResNet50 with different attentions (e.g., SE, CBAM, and HOGA) pretrained on ImageNet dataset as the backbone networks. We train the detection models on COCO dataset and take average precision as the evaluation metric [49]. More implementation details can be found in the supplementary material.

The results are summarized in Table 7. As can be seen, CBAM outperforms SE owing to the additional spatial attention. The model using AutoLA obtains the best score 27.78 AP, which is higher than the vanilla ResNet50 backbone by 2.77 AP with SSD framework. Compared with the manually designed SE and CBAM attentions, it also outperforms them by a large margin. Further, AutoLA also achieves better results in the Faster RCNN + FPN framework. It owes

Table 7: Comparison of object detection results on COCO dataset (Average Precision). We adopt SSD and Faster RCNN + FPN detection frameworks and apply different attention modules to the base network.

| | SSD | Faster RCNN + FPN |
|---|---|---|
| ResNet50 | 25.01 | 36.03 |
| ResNet50 + SE | 26.05 | 36.47 |
| ResNet50 + CBAM | 26.63 | 36.55 |
| ResNet50 + AutoLA | **27.78** | **37.21** |

to the larger receptive fields of HOGA introduced by high order attention operations, which enables to produce discriminative attention proposals and capture multi-scale context. These results further confirms the superiority of HOGA over existing attentions for object detection.

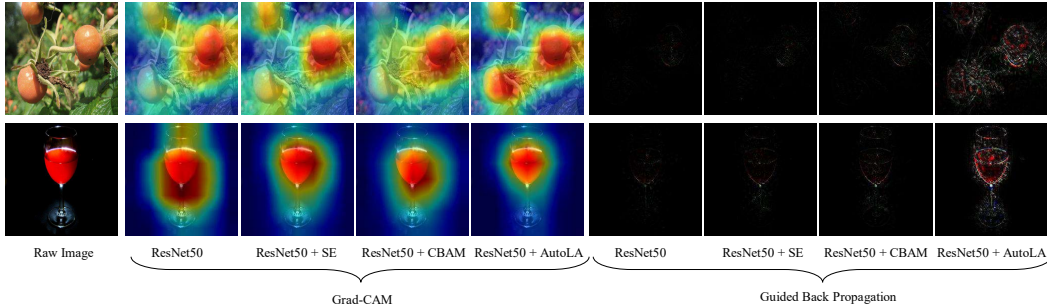

Raw Image    ResNet50    ResNet50 + SE    ResNet50 + CBAM    ResNet50 + AutoLA    ResNet50    ResNet50 + SE    ResNet50 + CBAM    ResNet50 + AutoLA

Grad-CAM        Guided Back Propagation

Figure 4: Visual inspection of the networks with Grad-CAM [52] and Guided Back Propagation [53]

## 4.7 Human Keypoint Detection Results on COCO

We further assess the generalization of AutoLA on the task of human keypoint detection which aims at detecting human body keypoints. We adopt the model in [50] and follow [51] for evaluating different attentions. Similar to Section 4.6, we plug different attention modules into ResNet50 and train them on ImageNet dataset. Then, we use them as the pretrained backbone networks and train them on COCO dataset. More implementation details can be found in the supplementary material.

The results are summarized in Table 8. As can be seen, CBAM improves the vanilla ResNet50 but does not perform better than SE for this task. We suspect that since the input of the keypoint detection model is a cropped and re-scaled person detection region where the human body is salient, therefore, the spatial attention may not benefit the model more given the channel attention. However, the searched HOGA outperforms all of them by large margins, demonstrating that diverse combinations and transformations of various attention operations actually matters.

Table 8: Human keypoint detection results

|  | AP | $AP^{@.5}$ | AR |
|---|---|---|---|
| ResNet50 | 72.8 | 89.9 | 78.5 |
| ResNet50 + SE | 73.6 | 90.2 | 79.3 |
| ResNet50 + CBAM | 73.6 | 90.0 | 79.3 |
| ResNet50 + AutoLA | **74.6** | **90.5** | **80.0** |

## 4.8 Visual Inspection on the Networks with Different Attention Modules

For the qualitative analysis, we apply the Grad-CAM [52] and guided back propagation [53] to inspect the "layer4" in ResNet50 with different attention modules. In Figure 4, we can see that the Grad-CAM masks of the network with AutoLA cover the target object regions more precisely than other methods. These results show that the network integrated with the searched HOGA can learn more discriminative features by attending to the target object and discarding irrelevant information.

Further, we use the class selectivity index metric [54] to analyze the features in different layers of models with different attention modules on the validation data of ImageNet. We also analyze the performance of CIFAR10 classification by the ResNet20, ResNet32, and ResNet56 backbones with different attention modules using Barnes-Hut-SNE [55]. These two visual inspections on different attention modules are shown and analysed in the supplementary material.

## 5 Conclusion and Future Work

In this work, we present the first attempt to search efficient and effective plug-and-play high order attention modules for various well-established backbone networks. We propose a new attention module named high order group attention and search its explicit architecture via a differential method efficiently. The searched attention module generalizes well on various backbones and outperforms manually designed attentions on many typical computer vision tasks. For the future work, we will formulate the backbone and attention architecture into a unified framework and search their optimal architectures in an alternative or synchronous manner.

**Acknowledgement** This work was supported by the the National Natural Science Foundation of China under grants 61771397, China Scholarship Council, Science and Technology Innovation Committee of Shenzhen Municipality under Grant JCYJ20180306171334997 and Australian Research Council Project FL-170100117.

**Broader Impact**

Machine learning and related technologies have already achieved remarkable performance in many areas. Current methods still require intensive empirical efforts for network design and hyperparameter fine-tuning. Our research can search the optimal high order group attention module automatically, and the searched module is computationally efficient and generalizes well to various tasks. It will help to build a strong deep neural network model automatically without having to rely on the manual design of the attention architecture. Since machine learning can promote the development of industry, healthcare, and education, AutoML can accelerate this process by offering various specific optimal models that fit different hardware platforms and latency constraints.

However, AutoML usually searches the model without domain knowledge, and may result in some uncertain and unreliable models that will make confusing decisions. The excessive trust in these decision will lead to many ethics issues. For example, when the diagnostic system optimized by AutoML leads to the death of the patients or other property damage, who should be responsible for this? What's more, the abuse of AutoML may cause horrible disasters, especially in military applications. Machine learning can optimize the design of weapons to make them adapted to the specific operational conditions. AutoML will speed up this process and makes it possible to search the optimal system under any different constraints and to customize the mass production of weapons. The weapon design systems optimized by AutoML will cause a great threat to world peace, and we advocate the AutoML will not apply to the field of military or warfare.

Further, AutoML will tip the scales in favor of the developed countries that are developing these technologies to improve living conditions across the board in a variety of ways. However, the labor force in developing countries is largely unskilled, and the use of AutoML in many cases means higher unemployment, lower-income, and more social unrest. The purpose of artificial intelligence in this condition should be to enhance their workforce skills, not to replace them. As a researcher, we need to work principally to make sure technology matches our values.

## Footnotes

*indicates equal contribution.

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
