[Supplementary Material]

# Auto Learning Attention: Supplementary Material

**Benteng Ma**[1,2*]          **Jing Zhang**[2] *          **Yong Xia**[1,3]          **Dacheng Tao**[2]

[1]Northwestern Polytechnical University, China
[2]The University of Sydney, Australia
[3]Research & Development Institute of Northwestern Polytechnical University, Shenzhen

{mabenteng@mail,yxia@}.nwpu.edu.cn
{jing.zhang1, dacheng.tao}@sydney.edu.au

## 1    Details on Network Training

**Image Classification**. For **CIFAR10**, we evaluate the backbones including ResNet20, ResNet32, and ResNet56 by deploying different attention modules [1]. We use the SGD optimizer to train each model for 500 epochs with the following same hyper-parameters. The initial learning rate is 0.1, and the momentum is 0.9. The weight decay is set as 0.0005. The batch size is 256.

For **CIFAR100**, we expand the base width (base channel number) from 16 to 64 compared with the models used in CIFAR10. Accordingly, the batch size is set as 128, and the other setups are the same with those in CIFAR10, including learning rate, momentum, weight decay, and the number of epochs.

For **ImageNet**, we evaluate the backbones, including ResNet18, ResNet34, ResNet50, ResNet101, and WiderResNet. The results are summarised in Table 3 of the paper. The input size is $224 \times 224$. We adopt the same data augmentation scheme as [2] for training. The learning rate starts from 0.1 and is divided by 10 every 30 epochs. We plug the searched HOGA from the above experiment into various backbone networks and train them for 100 epochs.

**Object Detection.** The input size is fixed as $300 \times 300$. The backbone used in the original **SSD** [3] is VGG16 [4]. We replace it with ResNet50 to evaluate the performance of different attention modules. Following [5], we modify the ResNet50 backbone accordingly. The conv5_x, average pooling, fc, and the softmax layers are removed from the original classification model. All strides in conv4_x are set to 1. Detector heads are enhanced by additional BatchNorm layers after each convolution. Additionally, we remove weight decay on every bias parameter and all the Batch Norm layers followed [6]. Then, SE, CBAM, and the searched HOGA are inserted into the ResNet50 backbone of SSD, respectively. We train the models for 65 epochs with the following setting. The batch size is 128. We use the SGD optimizer with an initial learning rate 0.01, which is divided by 10 at the 43 and 54 epochs. The momentum is 0.9. The weight decay is set as 0 for Bach Norm and biases, and 0.0005 for other layers. We also use linear warmup of the learning rate during the first epoch [7] for all the models.

For the **Faster RCNN + FPN** model, the base backbone is ResNet50. And the SE, CBAM, and AutoLA are inserted into ResNet50 backbone of Faster RCNN, respectively. For the detection models with different attention modules, we use FPN neck, RPN Head, and ROI Head by following [8]. All the parameters are the same with  [8]. We train the models for 14 epochs with the following setting. The batch size is 2. We use the SGD optimizer with an initial learning rate 0.01, which is divided by 10 at the 8 and 11 epochs. The momentum is 0.9 and the weight decay is 0.0001. We also use linear warmup of the learning rate during the first 500 iterations for all the models.

**Human Keypoint Detection.** The base model used for human keypoint detection is from [9], and the backbone is ResNet50. ResNet50 with different attention modules are implemented under the same setting. In the training stage, we use the Adam optimizer to train each model for 140 epochs with a batch size 128. The base learning rate is 0.001 and is divided by 10 at 90 and 120 epochs. Following [10], sub-pixel refinement tricks are also used for all the models.

Figure 1: Class selectivity index [11]. Each figure indicates the class selectivity index distribution for features in ResNet50 with different attention modules at the corresponding layer, including layer1, layer2, layer3, and layer4. As depth increases, we observe that AutoLA exhibits less class selectivity than other attention module.

## 2 Class Selectivity Index of Networks with Different Attention Modules

AutoLA can improve the performance of a deep network for many vision tasks as demonstrated by the experiments in the paper. In this part, we would like to gain some insight into how the learned features may differ from those learned by the ResNet50 backbone. To this end, we use the class selectivity index metric [11] to analyze the features in different layers of these models on the validation data of ImageNet. For the feature map in each layer, the class selectivity index metric computes the difference between the highest class-conditional mean activity and the mean of all remaining class-conditional activities over a given data distribution. The measurement is normalized into $[0, 1]$, where '1' indicates that a filter only fires for a single class, and '0' indicates that the filter produced the same value for every class. The metric is of interest to comparing the effect of attention modules since it provides a measure of the degree that features are shared across classes.

We compute the class selectivity index for intermediate features from 'layer1', 'layer2', 'layer3' and 'layer4', (the layer is the same with 'stage' defined in [12]). The results of different models are presented in Figure 1. In the early stage ('layer1'), the distribution of class selectivity for ResNet50, SE, CBAM, and AutoLA appears to be matched closely. However, with the increase of depth, the distributions begin to separate. For example, the distributions for 'layer4' appear more distinct where AutoLA exhibits less class selectivity than others. The enhanced high order context information by AutoLA helps the model to recognize locally ambiguous patterns while networks without such an ability may tend to use more highly specialized feature channels to achieve the same goal.

## 3 Visual Inspection on Different Attention Modules

**Grad CAM and Guided Back Propagation.** Following the approach described in Section 4.7 of the paper, we offer more samples generated with Grad-CAM [13] and Guided Back Propagation [14] in Figure 3. We obtain the consistent results with those in Section 4.7 that Grad-CAM masks of the network with AutoLA cover the target object regions more precisely than other methods. These results show that the network integrated with the searched HOGA can learn more discriminative context by attending to the target object and discarding irrelevant information.

**Barnes-Hut-SNE Analysis.** We further analyze the performance of CIFAR10 classification by the ResNet20, ResNet32, and ResNet56 backbones with different attention modules using Barnes-Hut-SNE [15]. Barnes-Hut-SNE [15] is an embedding method that is widely used for the visualization of high-dimensional data in scatter plots. We generate the results with the feature maps from layer3 (ResNet20, ResNet32, and ResNet56 contain three layers/stages in total) and show them in Figure 2.

Figure 2: Barnes-Hut-SNE Analysis. The SNE results are generated from ResNet20, ResNet32, and ResNet56 backbones with different attention modules on the feature maps from layer3 in each network. Each color indicates a category in CIFAR10.

We can see that networks with the searched HOGA can learn more discriminative features than those with other attention modules.

Figure 3: Visual inspection on different networks using Grad-CAM [13] and Guided Back Propagation [14]. In each row, the left is the original input image. The middle four images are generated using Grad-CAM on the layer4 of each architecture. The right four images are generated using Guided Back Propagation accordingly.