[Reviews · NeurIPS 2020]

Review 1

Summary and Contributions: The paper proposes a NAS method of attention. Different from traditional NAS that focuses on searching connections and operations, this paper tries to search high order attention. By modifying DARTS, the proposed method could accomplish the search within 1hour. The results indeed show improvement upon various baselines on several tasks.

Strengths: 1. The idea is straightforward, and the results are solid and indeed surpass baselines 2. Searching attention is an unexplored area of NAS, the scope of the paper is novel. 3. The method indeed improves across different tasks, which shows the universality of the proposed method.

Weaknesses: 1. The search space is quite limited, important types of attention such as non-local cannot be incorporated into the search space. 2. The compared baselines are not fair. It is not clear whether the performance improvement comes from the group split and high order attention or the searched arch. The authors should provide 1) random search baseline 3) SE and CBAM baseline with group split and high order attention. 3. More ablation analyses on the choice of k are needed to demonstrate the balance of speed and performance. 4. The proposed method increases the FLOPS by about 20%. It is necessary to provide some comparisons with baselines under the same FLOPS by scaling the width of the network.

Correctness: Yes

Clarity: Mostly clear to me, with some abuse of notations. For example, the meaning of k in Eqn1 and 2 are different. It is better to separate group the node index explicitly.

Relation to Prior Work: Yes

Reproducibility: Yes

Additional Feedback: It is interesting to see the results on stronger backbone, such as ResNet101 and ResNet152, etc. Also for the detection experiments, the baselines of COCO are too low from current view. I suggest the authors to try it on better baseline such as RetinaNet/FasterRCNN with ResNet50/101-FPN. ================================ The rebuttal does not change my opinion on this paper, especially the ablation analyses show that the random baseline is indeed comparable with the proposed method, and the performance improvement largely comes from group splitting. Thus I recommend rejection on this paper.


Review 2

Summary and Contributions: This paper presents a neural architecture search approach to search for an attention module, which can be plugged into various backbone networks. It proposes a new attention module called high order group attention and efficiently searches for its architecture via a differential method. Experiments show that the searched attention module with various backbones and outperforms hand-designed attention on several vision tasks.

Strengths: - The idea of architecture search for attention module is novel. Researchers have actively explored Neural Architecture Search as well as attention modules, but this paper is the first attempt to combine them together. - The proposed high order attention module is interesting. Previous work has sufficiently explored first order and second order attention modules including spatial and channel-wise attention modules, but higher order attention modules have not been explored to the best of my knowledge. - The approach produces good results on ImageNet classification using ResNet 18/34/50 backbones.

Weaknesses: - In section 3.1, the logic of extending HOGA from second order is not consistent with the extension from first order to second order; i.e., second order attention creates one more intermediate state U compared to the first order attention module. However, from the second order to higher order attention module, although intermediate states U0, U1 … are created, they are only part of the intermediate feature (Concatenating them will form U with full channel resolution). In this way, it seems we could regard the higher order attention module as a special form of second order attention module. - The paper does not clearly explain the intuition as to why different channel groups should have different attention mechanisms; i.e., in what specific way the network can benefit from the proposed channel group specific attention module. - Experiments are not solid enough: 1. There are no ablation studies on the effect of parameter numbers, so it is not clear whether the performance gain is due to the proposed approach or additional parameters. 2. Although there is good performance on imageNet classification with ResNet50/34/18, there are no results with larger models like ResNet101/152. 3. There are no results using strong object detection frameworks; the current SSD framework is relatively weak (e.g. Faster RCNN would be a stronger, more standard approach); it is not clear whether the improvements would be retained with a stronger base framework. - The proposed approach requires larger FLOPS compared to baselines; i.e., any performance gain requires large computation overhead (this is particularly pronounced in Table 3). - In Table 3 shows ResNet32/56 but L222 refers to ResNet34/50, which is confusing.

Correctness: Yes, overall they are correct.

Clarity: Some parts are unclear, as listed under weaknesses.

Relation to Prior Work: Yes, the proposed work has two main related fields, attention mechanism and neural architecture search, and both are clearly discussed.

Reproducibility: Yes

Additional Feedback: The idea of architecture search on attention module is novel and interesting, and the paper shows good performance gains on Imagenet classification. However, there are issues with experiments and unclear explanations as stated under "weaknesses". Thus, currently I tend to borderline reject this paper. ===== Post-rebuttal comments: I read the other reviews and the rebuttal.  The rebuttal addressed some of my issues regarding the experiments, but did not adequately address my concerns regarding unclear explanations.  Therefore, I maintain my original "Marginally below the acceptance threshold" rating.


Review 3

Summary and Contributions: The paper considers searching for an attention module for visual recognition architectures. In particular, the authors define an attention module parametrization that parametrizes a large number of attention module variants (including SE and CBAM). They then search for the optimal structure using a DARTS variant while keeping rest of the network structure / backbone fixed. The resulting module achieves consistent improvements across datasets (CIFAR, ImageNet) and tasks (COCO detection and pose estimation).

Strengths: 1. The studied problem is interesting and relevant for the community. 2. Although the search is performed using a single backbone (ResNet-20) and a single dataset (CIFAR-10) the found module transfers to different architectures (e.g. ResNet-56), datasets (e.g. ImageNet), and tasks (e.g. COCO detection). 3. The found module achieves consistent and solid improvements (~1 point) across all tested datasets (CIFAR-10, CIFAR-100, ImageNet) and tasks (COCO detection and key points). 4. The baselines are reasonable (vanilla, SE, CBAM), the training settings are clearly explained (Appendix 1), and fair (same across methods). 5. Qualitative visualization results are nice (e.g. Figure 4)

Weaknesses: 1. It is not very clear how exactly is the attention module attached to the backbone ResNet-20 architecture when performing the search. How many attention modules are used? Where are they placed? After each block? After each stage? It would be good to clarify this. 2. Similar to above, it would be good to provide more details of how the attention modules are added to tested architectures. I assume they are added following the SE paper but would be good to clarify. 3. Related to above, how is the complexity of the added module controlled? Is there a tunable channel weight similar to SE? It would be good to clarify this. 4. In Table 3, the additional complexity of the found module is ~5-15% in terms of parameters and flops. It is not clear if this is actually negligible. Would be good to perform comparisons where the complexity matches more closely. 5. In Table 3, it seems that the gains are decreasing for larger models. It would be good to show results with larger and deeper models (ResNet-101 and ResNet-152) to see if the gains transfer. 6. Similar to above, it would be good to show results for different model types (e.g. ResNeXt or MobileNet) to see if the module transfer to different model types. All current experiments use ResNet models. 7. It would be good to discuss and report how the searched module affect the training time, inference time, and memory usage (compared to vanilla baselines and other attention modules). 8. It would be interesting to see the results of searching for the module using a different backbone (e.g. ResNet-56) or a different dataset (e.g. CIFAR-100) and compare both the performance and the resulting module. 9. The current search space for the attention module consists largely of existing attention operations as basic ops. It would be interesting to consider a richer / less specific set of operators.

Correctness: The method is generally well-evaluated. However, it would be good to address the relevant points from the weaknesses section (4-7).

Clarity: The paper is generally well-written. However, it would be good to address the relevant points from the weaknesses section (1-3).

Relation to Prior Work: The paper discusses prior work clearly.

Reproducibility: No

Additional Feedback: Minor: - L184: where does the 91 number come from? I believe COCO has 80 categories - Table 3: I assume ResNet-56 is a typo and should be ResNet-50 Updated review: I thank the authors for the response. The rebuttal does not address my concerns regarding clarity and addresses some of my concerns about empirical evaluation. Overall, I maintain my original recommendation.


Review 4

Summary and Contributions: The paper presents neural architecture search algorithm (DARTS) to determine attention modules for various backbone networks. In particular, the authors first define the high order group attention (HOGA) that can be represented as a directed acyclic graph (DAG) and present an attention search space which includes typical attentional operations such as SE and CBAM. The authors demonstrate that the searched attention module can be applied to various backbones as a plug-and-play component and outperforms previous attention modules for many vision tasks.

Strengths: It is the first attempt to apply NAS to automate the attention module design. The authors define the novel concept of High Order Group Attention (HOGA) that generalizes previous attention modules (for example, SE and CBAM can be seen as a first-order and a second-order attention, respectively. The experiments are solid and the proposed novel High Order Group Attention (HOGA) attention module showed better performances than the existing SOTA approaches (e.g., SE and CBAM) for various bench marking datasets.

Weaknesses: The experimental comparisons are mainly conducted on ResNet backbones (with a different depth only). However, it is well known that the feature representations can be different by the backbone types and the results can obviously change to the choice of backbone. To show that the searched attention module is robust to these backbone variances, it is necessary to apply the module to other backbones such as WideResNet, Inception, DenseNet, and ResNext.

Correctness: The idea of continuous relaxation (equation 13) to ease the NAS optimization process is correct and good.

Clarity: It is well written and easy to understand.

Relation to Prior Work: There are some important missing attention modules. It is necessary to compare the HOGA with the following recent attention modules. [1] GCNet: Non-local Networks Meet Squeeze-Excitation Networks and Beyond (ICCV 2019) [2] Attention Augmented Convolutional Networks (ICCV 2019) [3] Gather-Excite: Exploiting Feature Context in Convolutional Neural Networks (NeurIPS 2018)

Reproducibility: Yes

Additional Feedback:

[Author Response · NeurIPS 2020]

**Fair comparison and ablation study.** The results on CIFAR10 were listed in Table R1. **(1)** We increased the width of
the backbone networks for SE and CBAM (denoted by "Wide") so that their parameters and FLOPs match those of
AutoLA. It reveals that HOGA searched by AutoLA (k=4)) still outperforms SE and CBAM by a large margin. **(2)**
We further customized SE and CBAM using the group split operation (denoted by "HOG"), resulting in a specific
instantiation of HOGA (i.e., k=4). From Table R1, the performances of customized SE and CBAM (order=4) are
improved significantly compared to the vanilla ones, attributing to the superiority of the proposed HOGA scheme.
**(3)** The HOGA searched by AutoLA outperforms its randomly search counterparts (denoted by "Rand"). The results
by random search exceed SE and CBAM. These results validate the superiority of the proposed AutoLA methods,
including the new concept HOGA and the architecture search algorithm. We also presented the ablation study on the
number of group split (i.e., the hyper-parameter k). Less groups mean lower order of attentions in HOGA, leading to
inferior performance. We tested the generalization ability of HOGA searched on ResNet56 (denoted by "AutoLA_56")
on a new backbone, i.e., ResNet20. Although the results are inferior to the ones searched directly on ResNet20, this
HOGA still outperforms SE and CBAM.

Table R1: Experiments with fair settings of parameters and FLOPGs and ablation study results on CIFAR10.

| | Acc (%) | Param (M) | FLOPS (G) | | Acc (%) | Param (M) | FLOPS (G) |
|---|---|---|---|---|---|---|---|
| ResNet20 + SE | 92.30 | 0.29 | 0.04 | ResNet32 + SE | 93.16 | 0.49 | 0.07 |
| ResNet20 + CBAM | 92.81 | 0.3 | 0.04 | ResNet32 + CBAM | 93.47 | 0.49 | 0.07 |
| ResNet20_Wide + SE | 93.16 | 0.36 | 0.05 | ResNet32_Wide_SE | 94.08 | 0.62 | 0.09 |
| ResNet20_Wide + CBAM | 93.13 | 0.37 | 0.05 | ResNet32_Wide_CBAM | 93.92 | 0.63 | 0.09 |
| ResNet20 + HOG_SE (k=4) | 92.87 | 0.32 | 0.05 | ResNet32 + HOG_SE (k=4) | 93.62 | 0.54 | 0.09 |
| ResNet20 + HOG_CBAM (k=4) | 93.07 | 0.35 | 0.05 | ResNet32 + HOG_CBAM (k=4) | 93.72 | 0.56 | 0.09 |
| ResNet20 + AutoLA (k=2) | 93.18 | 0.33 | 0.05 | ResNet32 + AutoLA (k=2) | 93.81 | 0.49 | 0.09 |
| ResNet20 + AutoLA (k=4) | 93.38 | 0.34 | 0.05 | ResNet32 + AutoLA (k=4) | 94.33 | 0.52 | 0.09 |
| ResNet20 + AutoLA_56 (k=4) | 93.31 | 0.35 | 0.05 | ResNet32 + AutoLA_56 (k=4) | 94.18 | 0.57 | 0.09 |
| ResNet20 + Rand_HOGA (k=4) | 93.28 | 0.35 | 0.05 | ResNet32 + Rand_HOGA (k=4) | 94.15 | 0.59 | 0.09 |

**Comparison on different backbones.** We presented the results for different backbones suggested by reviewer #4
and #5, including ResNeXt and the one searched by PNAS (Progressive Neural Architecture Search, ECCV2018)
on CIFAR10 in Table R2. In Table 3 (in the submission), we reported the results on ImageNet with WiderResNet
(denoted by WResNet18). Both tables show that the HOGA searched by AutoLA outperforms other attention modules
on CIFAR10 and ImageNet when deployed on highly variable architectures including ResNet, ResNeXt, PNAS, and
WiderResNet, indicating the consistent superiority of the HOGA searched by AutoLA over previous attention methods.

Table R2: Comparison of different attention modules on ResNeXt and PNAS.

| | Acc (%) | Param (M) | FLOPS (G) | | Acc (%) | Param (M) | FLOPS (G) |
|---|---|---|---|---|---|---|---|
| ResNext | 94.76 | 1.71 | 0.28 | PNAS | 93.34 | 0.72 | 0.08 |
| ResNext_SE | 95.22 | 2.23 | 0.30 | PNAS_SE | 93.71 | 0.75 | 0.08 |
| ResNext_CBAM | 95.31 | 2.24 | 0.31 | PNAS_CBAM | 93.79 | 0.76 | 0.08 |
| ResNext_AutoLA | 95.67 | 2.35 | 0.41 | PNAS_AutoLA | 94.10 | 0.91 | 0.11 |

**Comparison with other attention modules on larger backbones.** As suggested by reviewer #5, we further compared
the HOGA searched by AutoLA with other attention modules including **(1)** GCNet (Non-local Networks Meet Squeeze-
Excitation Networks and Beyond, ICCV 2019); **(2)** AugAtt (Attention Augmented Convolutional Networks, ICCV
2019); and **(3)** GENet (Exploiting Feature Context in Convolutional Neural Networks, NeurIPS 2018). The results
were listed in Table R3. With comparable or even less parameters and FLOPs, the proposed AutoLA outperforms other
attention methods by a substantial margin. We also compared AutoLA with SE and CBAM on a larger backbone (e.g.,
ResNet101). The results in Table R3 suggest that AutoLA still outperforms other attention modules.

Table R3: Experiments results by different attentions on ImageNet.

| | Top-1 Error (%) | Param (M) | FLOPS (G) | | Top-1 Error (%) | Param (M) | FLOPS (G) |
|---|---|---|---|---|---|---|---|
| ResNet50 + GENet | 22.00 | 31.20 | 3.87 | ResNet101 | 23.38 | 44.55 | 7.57 |
| ResNet50 + AugAtt | 22.30 | 24.30 | 7.90 | ResNet101 + SE | 22.35 | 49.33 | 7.58 |
| ResNet50 + GCNet | 22.30 | 28.08 | 3.87 | ResNet101 + CBAM | 21.51 | 49.33 | 7.58 |
| ResNet50 + AutoLA | 21.77 | 29.39 | 4.73 | ResNet101 + AutoLA | 20.95 | 51.81 | 8.94 |

**Large object detection model**. We evaluated the performance of AutoLA and CBAM for object detection on the
COCO dataset by equipping them with a powerful detection framework, i.e., Faster RCNN, as suggested by reviewers
#2 and #3. We used ResNet50 as the backbone. The mAP values are ResNet50(29.1), CBAM(30.8), AutoLA(33.7).
**Other details. (1)** The search space of HOGA contains $10^8$ potential structures although it can be further enlarged by
including other types of attention operations such as non-local attention. Note that, since non-local is computationally
heavy, an efficient implementation or approximation is necessary. **(2)** We termed the concept of high-order group
attention as a series of cascaded attention operations. In this sense, the proposed HOGA has a higher order than CBAM.
The effectiveness of designing or searching high-order attentions is also validated by the results of customized SE
and CBAM (order=4)in Table R1, attributing to the strong representation capacity resulted from combining diverse
nonlinear transformations (i.e.., different types of attention operations) among channel groups.

[Meta-Review · NeurIPS 2020]

All referees agree that the paper introduces an interesting and valuable idea but three referees have indicated reject because of missing experimental comparisons. The rebuttal includes all the experiments requested by the reviewers, providing additional insights and more compelling evidence on the benefits of the proposed approach. The referees have no further comments. After discussing with the SAC, we believe that the contribution of this paper is relevant and the rebuttal has properly addressed the concerns raised by the referees. Hence, I recommend acceptance. However, please consider revising your paper to include the results shared in the rebuttal and the reviewers' remarks to improve the clarity of the paper.